# Sex Differences in the Energy System Contribution during Sprint Exercise in Speed-Power and Endurance Athletes

**DOI:** 10.3390/jcm13164812

**Published:** 2024-08-15

**Authors:** Damian Archacki, Jacek Zieliński, Monika Ciekot-Sołtysiak, Ewa Anna Zarębska, Krzysztof Kusy

**Affiliations:** Department of Athletics, Strength and Conditioning, Poznan University of Physical Education, Królowej Jadwigi Street 27/39, 61-871 Poznań, Poland; jzielinski@awf.poznan.pl (J.Z.); ciekot@awf.poznan.pl (M.C.-S.); zarebska@awf.poznan.pl (E.A.Z.); kusy@awf.poznan.pl (K.K.)

**Keywords:** glycolytic, phosphagen, aerobic, sprint, energy expenditure, sex differences

## Abstract

**Background/Objectives**: A high level of specific metabolic capacity is essential for maximal sprinting in both male and female athletes. Various factors dictate sex differences in maximal power production and energy utilization. This study aims to compare the contribution of energy systems between male and female athletes with similar sport-specific physiological adaptations during a 15-s sprint exercise. **Methods**: The endurance group consisted of 17 males (23 ± 7 y) and 17 females (20 ± 2 y). The speed-power group included 14 males (21.1 ± 2.6 y) and 14 females (20 ± 3 y). The contribution of phosphagen, glycolytic, and aerobic systems was determined using the three-component PCr-LA-O_2_ method. **Results**: Significant differences were observed in the energy expenditure for all systems and total energy expenditure between males and females in both groups (*p* = 0.001–0.013). The energy expenditure in kJ for individual systems (phosphagen–glycolytic–aerobic) was 35:25:7 vs. 20:16:5 in endurance males vs. female athletes, respectively. In the speed-power group, male athletes expended 33:37:6 kJ and female athletes expended 21:25:4 kJ, respectively. The percentage proportions did not differ between males and females in any system. The contribution of the phosphagen–glycolytic–aerobic systems was 52:37:11 vs. 48:39:13 in endurance male and female athletes, respectively. For speed-power males vs. female athletes, the proportions were 42:50:8 vs. 41:50:9, respectively. **Conclusions:** Despite the differences in body composition, mechanical output, and absolute energy expenditure, the energy system contribution appears to have a similar metabolic effect between male and female athletes engaged in sprint exercises with similar sport-related adaptations. The magnitude and profile of sex differences are related to sports discipline.

## 1. Introduction

The relatively limited intramuscular ATP (adenosine triphosphate) stores (~5 mmol per kg of wet muscle) result in insufficient sustainability of contractile activity for extended periods. If stored ATP were the sole energy source during all-out exercise, the activity could be sustained for less than 2 s [1]. To restore ATP, activation of other metabolic pathways is necessary, which can be achieved through three integrated energy systems [2]. During exercise, ATP resynthesis is contributed to by all pathways in both anaerobic and aerobic processes. However, the extent of their involvement varies depending on the type, intensity, and duration of the sports activity.

The phosphagen system (anaerobic) relies on the degradation of phosphocreatine to creatine and the subsequent rephosphorylation of adenosine diphosphate to ATP. Together with cellular ATP reserves, this system dominates and delivers immediate energy during the initial moments of sprinting. The main reactions within the phosphagen system are creatine kinase, adenylate kinase, and adenosine monophosphate deaminase. The glycolytic system (anaerobic) operates by replenishing ATP sourced from blood glucose and muscle glycogen. Its activity increases during exercises lasting longer than a few seconds, utilizing a non-mitochondrial pathway [3]. Glycolysis involves a sequence of reactions that can be segmented into two phases: (i) an initial “investment” phase, necessitating the consumption of two ATP molecules, and (ii) a subsequent “payoff” phase where a net generation of ATP molecules takes place [4]. The aerobic system facilitates ATP resynthesis through mitochondrial respiration in the presence of oxygen. Within this system, free fatty acids and glycogen serve as the primary fuel sources for replenishing energy via the oxidation of carbohydrates, lipids, and amino acids [2,5]. Since different exercises elicit distinct metabolic demands, each sport and exercise modality exhibits a specific utilization pattern of these energy systems [6,7,8,9]. Sprinting is essential in competitive sports, contributing significantly to both speed-power and endurance disciplines due to factors such as acceleration, overtaking, and finishing [2]. Thus, there is a necessity for high performance in maximal sprint exercise across various sports.

A number of factors directly or indirectly influence the contribution of energy systems during maximal sprint exercise and determine sex differences in producing maximal power. These are primarily physique and body composition [10], physiological [11], biochemical [12,13], and biomechanical [14] factors. Although many studies have been conducted on sex differences, few have directly compared the contribution of energy systems between the sexes during sprinting. In one study, the authors analyzed martial artists using the Wingate test [15]. Other researchers compared men and women in the 100 and 200 m sprints [7]. On one hand, it was indicated that no sex differences were reported in the percentage contribution of energy systems; on the other hand, significant differences were reported in the total energy cost (kJ) between the sexes. Contrastingly, one study showed significant differences both in absolute and relative values in the comparison of males and females in repeated sprint tests [16]. The reports are not conclusive, and it is essential to note that the same sports adaptation and the lack of use of body descriptors, such as skeletal muscle mass (SMM) or leg lean mass (LLM), might potentially impact the outcome and further interpretations.

To our best knowledge, no one directly considered sex differences in energy system contribution during sprinting in the context of the specificity of the disciplines. Such data may help determine whether the quality of training stimuli should be diversified by sex. In addition, coaches and sports scientists may be provided with insights into the specific athletic predispositions of male and female athletes and help them select appropriate sports disciplines or specialties. Therefore, the present study aimed to compare the contribution of energy systems to sprinting exercise between male and female athletes practicing the same type of sport: speed and power or endurance. We hypothesize that there are sex differences in the contribution of the energy systems during sprints and that they are affected by sports specialization.

## 2. Materials and Methods

### 2.1. Participants

Initially, this study incorporated a cohort of 102 athletes representing diverse disciplines who underwent examinations during the general preparation period. After obtaining the data, male and female athletes were carefully selected to meet the criterion of a similar value of the peak and mean power per kilogram of skeletal muscle mass within each athletic group. Finally, this study included 62 male and female athletes, divided into two groups of different sports specializations. The detailed characteristics are presented in Table 1 and addressed in the Results section. The examined groups represented distinctively different metabolic and physiological profiles and had the same training status. The endurance group consisted of long-distance runners (specializing in 5 and 10 km) and triathletes (specializing in 1.5 km swimming, 40 km cycling, and 10 km running), whose competitive effort mainly involved low- to moderate-intensity activity. The speed-power group comprised sprinters (competing in the 60–200 m distances) and Olympic taekwondo athletes, who predominantly engaged in high-intensity explosive activities during training and competition. Both groups competed at the national level.

### 2.2. Procedures

The tests were conducted immediately after the transition (detraining) phase in the general preparation of the annual training cycle of the athletes. A week before the main examination, athletes participated in a familiarization session to acquaint themselves with the procedures and testing protocols. This study took place in the Human Movement Laboratory of the Department of Athletics and Strength and Conditioning at the Poznan University of Physical Education. The project was granted by the Ethics Committee at the Poznan University of Medical Sciences on 9 September 2020 (decision no. 627/20), and the research was carried out according to the principles of the Helsinki Declaration. All participants were familiarized with the study’s aim, procedures, and potential risks and signed the informed consent. To ensure accurate test results, athletes were instructed to be fast before undergoing the body composition analysis and to avoid high-intensity or long-duration training sessions for at least 24–48 h before testing. The blood samples were obtained before and after the Wingate test. Throughout the survey, the room temperature was maintained at 20–21 °C. The procedures are shown in Figure 1 and described in the subsequent sections.

### 2.3. Body Composition Analysis

A digital stadiometer (SECA 285, Hamburg, Germany) was employed to measure height and weight. Body composition analysis was performed using the Lunar Prodigy Pro device (GE Healthcare, Madison, WI, USA) and enCORE v16 SP1 software through the Dual X-ray Absorptiometry (DXA) method. The DXA scans were obtained and analyzed according to the protocol proposed by Nana [17]. The calculation of the total-body SMM was followed by the method proposed by Kim [18].

### 2.4. Wingate Test

The Wingate test was carried out according to standardization guidelines [19] using the Cyclus 2 cycle ergometer (RBM elektronik-automation GmbH, Leipzig, Germany). The test began with a 5-min warm-up, wherein participants pedaled at their own pace with resistance loads of 25 and 50 W, interspersed with brief sprints lasting up to 5 s. Then, after a ~2-min rest period, athletes were instructed to cycle as fast as they could for 15 s. To commence the test, participants were required to have a blood lactate concentration below 3 mmol∙L^−1^. If the level was higher, the test start was postponed until the desired value was achieved. The resistance load was set at 0.085 and 0.075 kg per kg of body weight for male and female athletes, respectively [20]. Throughout the test, participants received verbal encouragement to perform at their best. After completing the test, participants remained seated for a minimum of 10 min. Peak power was defined as the highest mechanical power achieved in a 5-s segment of the test, while mean power was calculated by averaging the instantaneous power values over the 15-s test duration. The fatigue index, indicating the rate of power decline, was determined by subtracting the maximum and minimum output power values and dividing the difference by the time of power drop [21,22].

### 2.5. Oxygen Uptake and Lactate

The respiratory parameters were recorded by MetaMax 3B-R2, and the subsequent analysis was conducted using MetaSoft Studio Software 5.1.0 (both: Cortex Biophysik GmbH, Leipzig, Germany). Measurements were recorded 5 min before (to determine resting V̇O_2_), continuously during the test, and 10 min after its completion. For lactate concentration determination, blood samples were obtained from the fingertip at three specific intervals: at rest, after the warm-up phase, and every minute in the period between the 3rd and the 10th minute following the test. The analysis was based on the resting and peak post-exercise values. The Biosen C-line apparatus (EKF Diagnostic GmbH in Barleben, Germany) was used to measure lactate levels utilizing capillary whole blood samples.

### 2.6. Calculation of the Energy System Contribution

The PCr-LA-O_2_ method was used to evaluate the contributions of the aerobic, glycolytic, and phosphagen energy systems as described by Beneke and Bertuzzi [23,24,25]. Energy expenditure for these systems was quantified in kilojoules, assuming a caloric equivalent of 20.9 kJ per 1 L of O_2_. Energy derived from the aerobic system (E_AER_) was determined by subtracting the resting V̇O_2_ (5 min) from the V̇O_2_ recorded during maximal exercise, employing the trapezoidal method [23]. Energy from the glycolytic system (E_LA_) was assessed using the oxygen equivalent derived from blood lactate concentration. This was approximated as the difference between peak lactate concentration and resting lactate levels. A net value of 1 mmol∙L^−1^ was equated to 3 mL O_2_∙kg body mass^−1^ [26]. Energy from the phosphagen system (E_PCR_) was calculated based on the fast component of the excess post-exercise oxygen consumption (EPOC) [27]. The V̇O_2_ consumption data were collected during a 10-min recovery period after the test, following procedures outlined in the study [28], and calculated using the bi-exponential model [29]. The subsequent formulas were applied:V̇O_2(t)_ = V̇O_2baseline_ + A*_f_*[e^−(t–td)/τ^*_f_*] + A*_s_*[e^−(t–td)/τ^*_s_*]
E_PCR =_ A*_f_* ∙ τ*_f_*_,_
where V̇O_2(t)_ is the oxygen uptake at time *t*; V̇O_2baseline_ is the oxygen uptake at rest; A is the amplitude; td is the time delay; *τ* is the time constant; *f* denotes the fast; and *s* denotes the slow component of EPOC. To achieve an accurate fit for each EPOC kinetic curve, we conducted calculations at various time points of oxygen consumption, from the 3rd to the 10th minute of recovery. Due to consistently reliable outcomes, we specifically focused on the 7-min EPOC period in our analysis. This approach yielded high coefficients of determination for all the kinetic curves assessed (r^2^ = 0.80–0.98). Our approach to analyzing EPOC aligns with methodologies adopted in other studies, which typically involve assessing recovery time from the 6th to the 10th minute following various high-intensity exercises such as the 100 m sprint and 15- or 30-s all-out cycling efforts [30,31,32,33]. Furthermore, Bertuzzi et al. [28] argued that monitoring the initial 6 min of V̇O_2_ recovery post moderate- and high-intensity exercise adequately reveals the contribution of the phosphagen system. Additionally, we edited the raw data according to the methods proposed by Lamara and Myers [34,35]. Consequently, values exceeding 3 standard deviations were omitted to exclude deviations potentially caused by coughing, sighing, etc. The data underwent analysis using the GEDAE-LaB software (first version available on GitHub.com), developed by researchers from Sao Paulo University and the Federal University of Pernambuco, Brazil [28]. The total energy expenditure (E_TOT_) was calculated as follows:E_TOT_ = E_AER_ + E_LA_ + E_PCR_

The proportion of each energy system was calculated as a percentage of (E_TOT_).

### 2.7. Statistical Analyses

The normality of the data distribution was tested using the Shapiro–Wilk test. As the distribution was found to be normal, all values were shown as the mean and standard deviation. 

A multivariate analysis of variance (MANOVA) was performed to test the effects of group, sex, and their interaction on the set of dependent variables (energy from three systems). Following this, a two-way analysis of variance (ANOVA) was conducted for each dependent variable (energy system) to further examine the effects of group, sex, and their interaction. Also, an analysis of covariance (ANCOVA) was performed to account for the variables BF and SMM. In addition, a one-way ANOVA was conducted to assess the magnitude of the differences in descriptive variables among athletic groups between males and females. 

The Levene’s test was employed to evaluate the homogeneity of the variance. The Bonferroni post-hoc test was used in the presence of statistically significant main effects. The partial eta squared (*η*^2^) was computed as an indicator of the effect size and interpreted as small (0.2), medium (0.05), or large (0.8) [36]. All analyses were performed using Statistica software 13 (TIBCO Software Inc.,1984-2017, Palo Alto, Santa Clara, CA, USA).

## 3. Results

### 3.1. Body Composition Analysis and Skeletal Muscle Mass

Sex differences were observed within both athletic groups for all variables except for blood lactate concentration at rest. Male athletes from both groups expressed significantly greater height, body weight, SMM and LLM (both% and kg), and lower levels of body fat compared to female athletes within the group (Table 1). There were differences between athletic groups within the same sex: males from the speed-power group had significantly greater body mass, LLM (kg), and LA_PEAK_, and lower body fat (%) compared to males from the endurance group. Females from the speed-power group had higher body mass, body fat (both kg and %), SMM (kg), LLM (kg), and LA_PEAK_.

### 3.2. Energy Systems Contribution

The main results are presented in Table 2 and Figure 2, with detailed mean values of energy expenditure and percentage contribution in Table 3. We revealed a significant multivariate effect of the athletic group, sex, and their interaction on the energy expenditure (kJ) and percentage contribution of energy systems. 

Significant univariate main effects of sex on energy expenditure (kJ) within each energy system and the effect of sports discipline on the glycolytic system were observed (Table 2 and Figure 2A). Male athletes showed significantly greater absolute energy expenditure than females for each system and total energy expenditure within both athletic groups. Moreover, a significant effect of the athletic group emerged for the glycolytic system, where significant differences in energy expenditure were observed between male endurance and speed-power (*p* < 0.001) groups and between female endurance and speed-power groups (*p* = 0.004), similarly for total energy expenditure (*p* < 0.001). Both male and female athletes in the speed-power group exhibited significantly higher energy expenditure in the glycolytic system and total energy expenditure than athletes of the same sex in the endurance group. 

In contrast to absolute energy expenditure, there were no significant effects of sex on the percentage energy system contribution for any energy system (Table 2, Figure 2B). However, significant effects of the athletic group were shown for all energy systems. The endurance and speed-power male athletes differed significantly in the contribution of the phosphagen (*p* = 0.035) and glycolytic (*p* = 0.005) systems, whereas endurance and speed-power female athletes differed only in the glycolytic system (*p* = 0.030).

In addition, total body fat and skeletal muscle mass were included as covariates (Table 2). Once the covariates were introduced, the effect of gender on absolute energy expenditure disappeared (not significant for any of the systems), and the effect of sports group was strengthened (significant for all systems). This was associated with a significant muscle mass effect on the energy produced by the phosphagen and glycolytic systems. In the case of the percentage energy system contribution, significant statistical effects remained unchanged after the introduction of covariates, i.e., only the sport specialty differentiated the percentage proportions, and sex was still not a significant factor, nor were fat content and muscle mass.

### 3.3. Peak and Mean Power

Males achieved greater PP_LLM_ and MP_LMM_ than females in the speed-power group (*p* = 0.008). The endurance group showed no sex differences in any mechanical variables calculated per kilogram of SMM and LLM (Figure 3).

### 3.4. Size of the Sex Differences

Figure 4 shows the summary of sex differences in somatic and mechanical variables, and energy expenditure within the endurance and speed-power groups. The results were expressed as Cohen’s effect size and interpreted as small (0.2), medium (0.5), or large (0.8). 

Overall, the effect size in all somatic variables, i.e., SMM and LLM, was slightly greater among the endurance group than speed-power (0.63–0.79 vs. 0.37–0.62), respectively. In the absolute mechanical variables (PP and MP), the effect size was at a similar (medium) level in both groups between sexes (endurance: 0.76–0.77 vs. speed-power: 0.68–0.69). The effect size of differences in the relative mechanical variables (per kilogram of SMM) among the speed-power group was at a medium level in PP_LLM_, small in MP_LLM._, PP_SMM_, and MP_SMM_. In contrast, in the endurance group, PP_SMM_ and MP_SMM_ did not reach statistical significance (*p* = 0.130. and 0.103, respectively). Despite the significance of PP_LLM_ and MP_LLM_ (*p* = 0.039 and *p* = 0.037, respectively), the effect size was negligible. Energy expenditure for systems and total energy expenditure (kJ) presented a small effect in both groups, except for the aerobic system in the endurance group and the phosphagen system in the speed-power group, which showed a negligible effect.

## 4. Discussion

This is the first study that compared the response of the energy system contribution and the metabolic energy expenditure during sprint exercise in male and female athletes of different sports specializations with similar PP_SMM_ and MP_SMM_ during the 15-s Wingate sprint test. The key findings are that (i) male and female athletes significantly differ in absolute energy expenditure, but they do not differ significantly in the proportions of energy system contribution, and (ii) the size and profile of sex differences in factors related to the Wingate test performance depend to some extent on the type of sport. The latter observation may be related to the differences in metabolic adaptations and predispositions between the athletic groups. Endurance athletes typically exhibit superior efficiency in aerobic components [37], attributed to central factors such as stroke volume, cardiac output [38], and V̇O_2_ max [39], as well as peripheral factors including mitochondrial density [40,41]. In contrast, speed athletes generally demonstrate greater efficiency at the intramuscular level, primarily due to their higher reliance on type IIa fibers [42], which are intermediate and exhibit enhanced adaptability to fatigue [43].

### 4.1. Sex Differences in Body Composition and Peak Power Output

The absolute and percentage values of body composition strongly differentiated our male and female athletes from both groups, which is in agreement with other studies [36,37,38,39]. As for the most obvious differences, male athletes have a higher lean body mass, SMM, and lower levels of fat mass than females in the same sports discipline, which provides them with the ability to generate higher absolute power and achieve a higher power-to-weight ratio. Perez-Gomez et al. [44] demonstrated a direct association between the absolute lean mass of the lower limbs and peak power output during the Wingate test. On one hand, sex differences in muscle fibers are evident. Men generally exhibit larger cross-sectional areas across all types of muscle fibers, with a greater percentage and area distribution for type II, IIA, and IIX fibers [45]. Conversely, women demonstrate greater distribution and area for type I fibers [45]. Additionally, while type II fibers contract at higher velocities and generate more force and power, type I fibers have a greater capacity for oxidative energy production and exhibit more endurance [46,47,48].

However, it is important to note that nearly 90% of the subjects in the above-mentioned review by Nuzzo et al. were from the general population or individuals with various medical conditions, while only 11% of the data involved athletes. On the other hand, Costill et al. [49], who examined female and male track athletes, indicated that both sexes exhibit similar muscle fiber composition, selected enzyme levels, and percentages of slow-twitch fibers across various events, with the only difference being the larger fiber areas in male athletes. This supports earlier findings suggesting that an athlete’s success in strength, speed, and endurance events is partly due to their genetic potential [50]. This observation may account for the lack of significant sex differences in energy system contribution and even in absolute energy expenditure after introducing SMM as a covariate, as observed in our study. If men and women engaged in high-performance sports exhibit comparable fiber properties and enzyme levels, it is plausible to expect a similar proportion of energy system activity between the sexes. Consequently, analogous structural and functional resources might result in similar energy contributions. We revealed that the profiles of sex differences in body components seem to be discipline-dependent, with stronger differences in endurance than in the speed-power group (Figure 4). Similarly, one study that examined male and female long-distance runners and sprinters indicated that the differences between sexes in somatic factors related to maximum oxygen uptake were greater in endurance athletes [51]. 

The sex differences in absolute power were similar in both groups. Overall, in line with our results, Hallam and Amorim [52] showed that the sex differences in sports performance among the top 20 athletes were consistently smaller in sprinters (~10%) than in middle- and long-distance runners (~15%) over the years. 

However, after the calculation of the mechanical variables per kilogram of SMM and LLM, the differences were drastically reduced in the endurance group (nonsignificant or negligible effect). The speed-power of male and female athletes still differed significantly, showing only a small effect in PP_SMM_, MP_SMM_, MP_LLM_, and medium in the PP_LLM_. On one hand, some authors showed that expressing the peak or mean power calculated for a body descriptor such as SMM or lean body mass significantly reduced sex differences [53]. On the other hand, several studies have demonstrated consistent differences even after normalization for body dimensions [54,55,56]. Nonetheless, to achieve greater reliability, predictability, and comparability when assessing individuals of different sexes, it is necessary to utilize body descriptors. In cases where a subject performs the test in a seated position, it may be even more suitable to use the mass of the lower limbs, e.g., leg lean mass, as a descriptor [57].

In our study, the differences in the speed-power disciplines may be greater due to the specificity of sports training and performance, which rely to a greater extent on producing higher maximal power output (mainly through the lower limbs) than middle- and long-distance activity. This may be explained by the fact that the muscle mass in the lower limbs is considered a main factor in determining sex differences in peak power during the Wingate test [44,57]. This is strictly associated with force production [58]. Hallam and Amorim [52], in turn, suggest that the variance between sexes among endurance runners is mainly related to the differences in maximal oxygen consumption. Analogously, the short duration of the Wingate test might be associated with greater intra-group discrepancies in speed-power athletes.

In general, sex differences in the mechanical variables are more visible among untrained subjects or participants without long-term sports adaptation [44,53,54]. Additionally, Serresse et al. [56] examined physical education students, sedentary individuals, and athletes from various disciplines, indicating that sex differences in their performance exist, but they seem to diminish in trained subjects. Our research and that of other authors need not contradict each other. It seems that the type of specific exercise, testing protocol (sprint or endurance), and long-term sport adaptation account for the size of the sex differences in mechanical variables.

### 4.2. Sex Differences in Energy Expenditure and Energy System Contribution

Significant sex differences were observed in the absolute values (kJ) for phosphagen, glycolytic, oxidative, and total energy expenditure in both groups (Figure 3). Nevertheless, the percentage contribution of energy systems (Figure 1) and total energy expenditure calculated per kilogram of SMM did not differ in any group (Table 3). This is in line with the study of Tortu et al. [15,59] and Duffield et al. [7], who showed significant differences between sexes in the total energy cost but not in the energy system contribution (%).

On one hand, in general, males tend to have a greater ability to produce the necessary ATP to sustain muscle contraction during a high-intensity sprint due to their higher anaerobic power, which may result in a greater amount of energy that derives from the phosphagen pathway [44]. Sex differences are also evident in the anaerobic capacity and the activity of glycolytic enzymes such as phosphofructokinase (PFK), with men demonstrating ~15–29% higher activity and lactate dehydrogenase (LDK) in comparison to women, as reported in various studies [53,60,61,62]. On the other hand, it has been proven that the sex differences in the activity of LDK or PFK can be eliminated by normalizing the muscle cross-sectional area [62].

It seems that the differences in energy expenditure among our athletes primarily stem from variations in their SMM. This is evidenced by the fact that the sex effect disappeared after introducing the SMM covariate. Marra et al. [63] indicate that when SMM is accounted for, the sex-based differences in energy expenditure diminish, supporting the idea that SMM is a critical factor in determining energy requirements and expenditure rather than sex alone. However, their ATP production, enzyme activity, and metabolic response remain consistent. Hallam and Amorim [52] propose that the sex gap widens at lower performance levels. Massini et al. [64] indicated no differences between male and female swimmers in the contribution of energy systems in the distances 50 and 100 m. This is also in line with the study of Almeida et al. [65], who suggested that energetics (%) during short- and middle-distance swimming did not differ between sexes per unit of body mass. Yang et al. [66] showed significant sex differences among recreationally active subjects during the graded incremental exercise test in energetic contributions, but the athletes did not differ within the group. Hence, the absence of specific adaptation could have fueled the sex disparities observed in previous studies [53,54]. According to Zamparo et al. [67], the energy cost at submaximal swimming speed is approximately 20–30% lower in females than in males. However, when these values are normalized for body size, the differences disappear. Similarly, the variability in the energy cost of kayaking per unit distance may be attributed to sex differences, but these differences tend to diminish in well-skilled and homogeneous groups [68].

Additionally, in our study, the size of the sex differences within athletic groups in the energy expenditure for the phosphagen system was greater in the endurance group than in the speed-power group. This observation may be attributed to their training specificity, which in the speed-power group demands similar engagement from both sexes within this metabolic pathway and results in smaller discrepancies in their dominating system. Conversely, endurance athletes rely to a lesser extent on the phosphagen system in their training regime, allowing physiological and biomechanical factors that differentiate between sexes to exert greater influence, consequently leading to increased variability. In contrast, the effect size on energy expenditure for the aerobic system was greater within the speed-power group than in the endurance group. Both athletic groups presented almost equal small effect sizes in the glycolytic energy expenditure. 

Nevertheless, despite the reported differences, the effect sizes on energy expenditure were small or negligible in both groups. Perhaps, if the ongoing long-term adaptation exists, the sex differences in the contribution of energy systems tend to decrease. Perhaps, in the general population, sex may play a more important role in energy system contribution due to greater variability among subjects with no specific adaptation [66].

Generally, despite the differences in the somatic and mechanical variables, athletes with specific adaptations exhibit equal contributions to energy systems. Therefore, from a practical standpoint, organizing the training regimen similarly can yield equivalent metabolic reactions in athletes with the same specific adaptations. It is the athlete’s sport specialty, with its associated metabolic predispositions and adaptations, that more strongly determines the percentage contribution of energy systems, not the biological sex of the athlete itself. Therefore, if male and female athletes are at the same sports level, we should expect the same proportion of these systems in maximal all-out exercise. In turn, the obvious sex differences in absolute energy expenditure during sprinting are mainly due to differences in skeletal muscle mass.

### 4.3. Limitations

The data obtained in our study are limited to two groups of athletes with different adaptations (endurance vs. speed-power) during the preparation phase of their annual training cycle. Consequently, these results cannot be extrapolated to untrained individuals lacking long-term training adaptations. Future research should consider including untrained subjects as well as athletes in various phases of the annual cycle, such as transition and competition periods, to examine changes or the absence thereof in the contribution of energy systems.

## 5. Conclusions

The novelty of the study is the comparison of sex differences within two distinct sport-related physiological adaptations in athletes with the same training status and performance level. While the body composition and mechanical output are different, the energy system contribution is almost the same in sprint exercise among males and females of the same sport-related adaptation. Therefore, despite the differences in absolute energy expenditure, the metabolic effect on the energy system contribution is similar among male and female athletes. The magnitude and profile of sex differences in somatic and mechanical variables related to Wingate test performance are affected by sports discipline. Our findings suggest that training and performance strategies aimed at energy metabolism during sprinting should consider the specific demands of the sport and individual body composition (in particular muscle mass) rather than relying on sex-based generalizations.

## Figures and Tables

**Figure 1 jcm-13-04812-f001:**
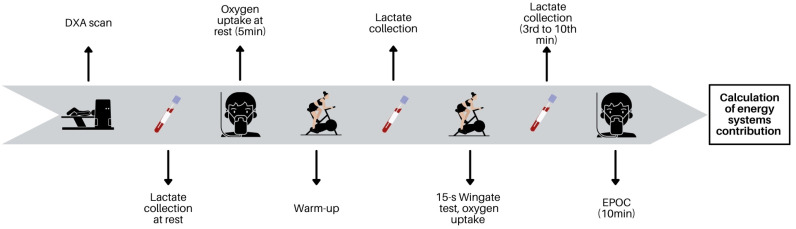
Testing procedure. The graph was created using Canva.

**Figure 2 jcm-13-04812-f002:**
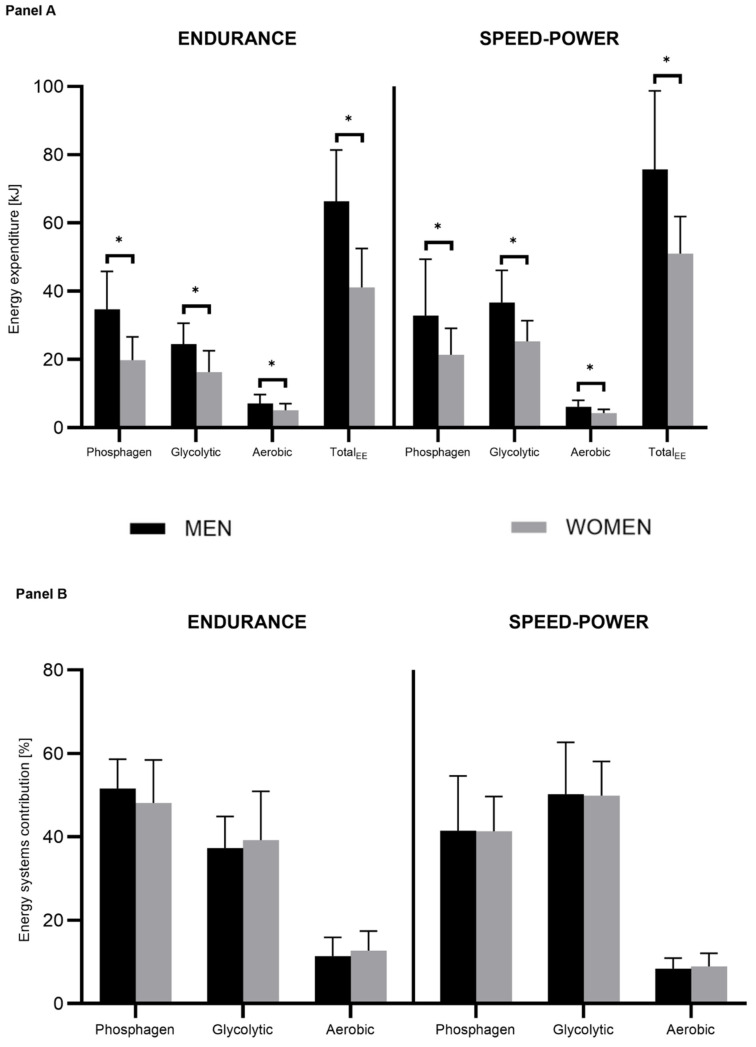
Energy system contribution (panel **A**) and energy expenditure (panel **B**) during the 15-s Wingate test among males (black bars) and females (gray bars) in endurance and speed-power groups. * Significant differences between sexes.

**Figure 3 jcm-13-04812-f003:**
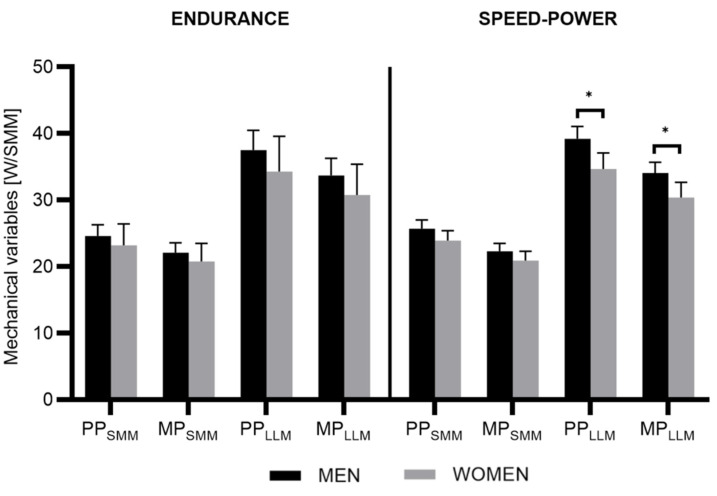
The comparison in mechanical variables calculated per kilogram of skeletal muscle mass among males and females in the endurance and speed-power groups. * Significant differences between sexes.

**Figure 4 jcm-13-04812-f004:**
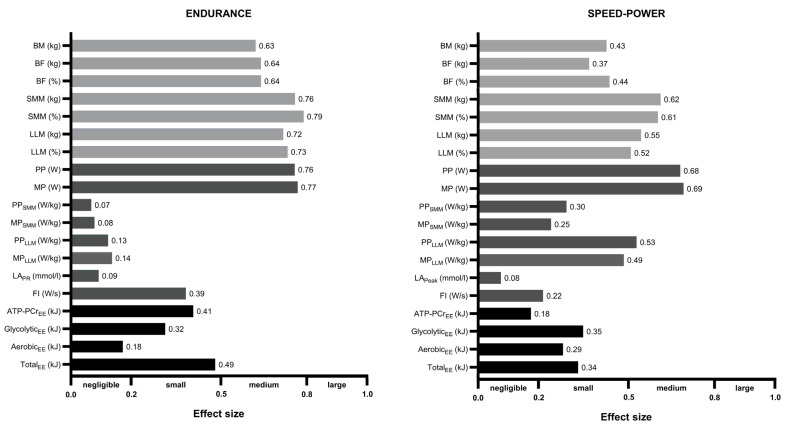
The effect size was significant for differences in body composition, mechanical variables, and energy expenditure between males and females in the endurance and speed-power groups.

**Table 1 jcm-13-04812-t001:** The basic characteristics of the athletic groups and the differences between male and female athletes.

	Endurance		Speed-Power	
	Male	Female	Male	Female
Age (y)	23 ± 7	20 ± 2	21.1 ± 2.6	20 ± 3
Height (cm)	179 ± 6 ^#^	165.1 ± 4.2	182.1 ± 5.4 ^#^	171.1 ± 9
BM (kg)	68 ± 6.6 ^#^*	54 ± 4.4 ^†^	78 ± 9.4 ^#^	62.9 ± 11
BF (kg)	9.0 ± 2.5	11.7 ± 1.8 ^†^	12.4 ± 4.0	15.8 ± 5.9
BF (%)	13.2 ± 3 ^#^*	21.8 ± 3.6 ^†^	15.9 ± 4.8 ^#^	24.5 ± 5
SMM (kg)	29.9 ± 3.2 ^#^*	20.3 ± 2.3 ^†^	34.6 ± 5 ^#^	24 ± 3.6
SMM (%)	43.9 ± 1.4 ^#^	37.6 ± 1.9	44.2 ± 2.8 ^#^	38.3 ± 2.1
LLM (kg)	19.6 ± 2.1 ^#^*	13.8 ± 1.7 ^†^	22.6 ± 3.1 ^#^	16.5 ± 3
LLM (%)	81.7 ± 3 ^#^	70.5 ± 4	79.7 ± 5 ^#^	69.5 ± 5
LA_REST_ (mmol/L)	2.2 ± 0.6	2.1 ± 0.6	2.3 ± 0.6	2.2 ± 0.5
LA_PEAK_ (mmol/L)	7.8 ± 1.1 ^#^*	6.9 ± 1.8 ^†^	9.6 ± 1.7	8.7 ± 1.6

Values are expressed as means ± standard deviations (*p* < 0.05). Abbreviations: BM—body mass; BF—body fat; SMM—skeletal muscle mass; LLM—leg lean mass; LA_REST_—lactate concentration at rest; and LA_PEAK_—lactate concentration at peak. # Significantly different from female athletes within the same group. * Significantly different between male athletic groups. † Significantly different between female athletic groups.

**Table 2 jcm-13-04812-t002:** The main effects of sex and sports discipline on absolute energy expenditure and percentage energy system contribution during 15-s all-out cycling. Statistical significance (*p*-value) and effect sizes (η*^2^*) for multivariate analysis of variance (MANOVA), univariate two-way analysis of variance (ANOVA), and analysis of covariance (ANCOVA) with total body fat and skeletal muscle mass as covariants are presented. Significant effects are marked with an asterisk (*).

	MANOVA					
	Energy expenditure					
Sex	*p* = 0.011 *, η^2^ = 0.210					
Discipline	*p* = 0.001 *, η^2^ = 0.626					
Sex ∗ Discipline	*p* = 0.027 *, η^2^ = 0.187					
	Energy system contribution					
Sex	*p* = 0.003 *, η^2^ = 0.241					
Discipline	*p* < 0.001 *, η^2^ = 0.949					
Sex ∗ Discipline	*p* = 0.007 *, η^2^ = 0.221					
	Phosphagen		Glycolytic		Aerobic	
	*p*-value	η^2^	*p*-value	η^2^	*p*-value	η^2^
	ANOVA					
	Energy expenditure					
Sex	<0.001 *	0.278 *	<0.001 *	0.339 *	<0.001 *	0.206 *
Discipline	0.974	0.000	<0.001 *	0.377 *	0.076	0.052
Sex ∗ Discipline	0.545	0.006	0.385	0.013	0.875	0.000
	Energy system contribution					
Sex	0.466	0.009	0.754	0.001	0.309	0.017
Discipline	0.001 *	0.165 *	<0.001 *	0.261	0.002 *	0.148
Sex ∗ Discipline	0.516	0.007	0.682	0.002	0.573	0.005
	ANCOVA					
	Energy expenditure					
Sex	0.482	0.009	0.169	0.033	0.584	0.005
Discipline	<0.001 *	0.521	0.001 *	0.190	<0.001 *	0.349
Body fat	0.488	0.008	0.461	0.010	0.274	0.020
Muscle mass	0.017 *	0.096	0.008 *	0.117	0.709	0.002
	Energy system contribution					
Sex	0.371	0.014	0.315	0.017	0.103	0.046
Discipline	<0.001 *	0.776	<0.001 *	0.429	<0.001 *	0.425
Body fat	0.982	<0.001	0.995	<0.001	0.953	<0.001
Muscle mass	0.607	0.004	0.652	0.004	0.961	<0.001

**Table 3 jcm-13-04812-t003:** Energy system contribution and energy expenditure among males and females in endurance and speed-power groups.

	Endurance		Speed-Power	
	Male	Female	Male	Female
	Relative values (%)			
E_PCR_	51.6 ± 7 *	48.1 ± 10.3	41.5 ± 13.1	41.3 ± 8.4
E_LA_	37.3 ± 7.6 *	39.2 ± 11.7 ^†^	50.2 ± 12.5	49.9 ± 8.2
E_AER_	11.1 ± 4.5	12.7 ± 4.7	8.3 ± 2.5	8.8 ± 3.5
	Absolute values (kJ)			
E_PCR_	34.7 ± 11.1 ^#^	19.8 ± 6.8	32.9 ± 16.5 ^#^	21.4 ± 7.7
E_LA_	24.5 ± 6.1 ^#^*	16.3 ± 6.2 ^†^	36.7 ± 9.4 ^#^	25.3 ± 6.1
E_AER_	7.1 ± 2.6^#^	5.1 ± 1.9	6.1 ± 1.9 ^#^	4.3 ± 1.1
	Total energy expenditure			
EE_TOTAL[kJ]_	66.4 ± 15 ^#^*	41.1 ± 11.4 ^†^	75.7 ± 23 ^#^	51.0 ± 10.9
EE_TOTAL/BM [kJ/kg]_	0.98 ± 0.21	0.76 ± 0.21	0.97 ± 0.25	0.81 ± 0.14
EE_TOTAL/SMM [kJ/kg]_	2.23 ± 0.48	2.04 ± 0.58	2.18 ± 0.48	2.13 ± 0.36

Values are expressed as means ± and standard deviations (*p* < 0.05). # Significantly different from female athletes within the same group. * Significantly different from male athletes between groups. † Significantly different from female athletes between groups. Legend: E_PCR_—phosphagen system; E_LA_—glycolytic system; E_AER_—aerobic system; EE_TOTAL_—total energy expenditure; EE_TOTAL/BM_-energy expenditure/body mass; EE_TOTAL/SMM_—energy expenditure/skeletal muscle mass.

## Data Availability

The raw data supporting the conclusions of this article will be made available by the authors on request.

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
