# Peer review of "Sex Differences in the Energy System Contribution during Sprint Exercise in Speed-Power and Endurance Athletes"

_jcm, 2024, doi:10.3390/jcm13164812_

Round 1
Reviewer 1 Report (Previous Reviewer 1)
Comments and Suggestions for Authors
Please see the attachment

Author Response
Thank you for reviewing our manuscript. Detailed responses and the corresponding revisions, highlighted in track changes, can be found in the re-submitted files below.
Comment 1:
Please put references in the same bracket.
Response 1:
Thank you for your suggestion. All the multiple references have been put in the same bracket.
Comment 2:
In the introduction, it is recommended not to use references in this format. I suggest describing the subject and referencing it at the end.
Response 2:
Thank you for the feedback.
Done.
Comment 3:
There was a lack of description and justification of the reason for this study, demonstrating its importance. Emphasizing the gaps on the subject.
Response 3:
Thank you for your valuable feedback. We appreciate your observation and added a brief justification in the introduction.
“To our best knowledge, no one directly considered sex differences in energy system contribution during sprinting in the context of the specificity of the disciplines. Such data may help to determine whether the quality of training stimuli should be diversified by sex. In addition, coaches and sports scientists may be provided with insights into the specific athletic predispositions of male and female athletes and help them select appropriate sports discipline or specialty.”
Comment 4:
Please put this sample characteristic data into a table.
Response 4:
Thank you for the comment. The data have been moved to Table 1 in the Results section.
Comment 5:
I suggest putting the experimental design in illustrative format, in a figure.
Response 5:
Thank you for your suggestion. The experimental design has been presented in the new Figure 1.
Comment 6:
This is an erroneous pattern of tests applied in research. Because 5 minutes of warm-up (mainly at 25 to 50w) is not enough to physiologically activate the body to submit to stimuli, especially at high intensity.
Response 6:
Thank you for your feedback. We appreciate your valuable insights on the warm-up protocol.
Obviously, a warm-up of constant low-to-moderate intensity would be inappropriate. We based our protocol on the work of Beneke et al. (2002) and other authors who employed a similar approach before the Wingate test (Lovell et al. 2012, Doria et al. 2020). A key aspect of our warm-up was incorporating short sprints to replicate the specificity of the main task. Unlike Beneke, who recommended only two sprints of 3 seconds each, our athletes performed additional sprints up to 5 seconds, so the pre-adaptation was even more pronounced. When developing or warming up for maximal speed-power/velocity, a relatively low external load must be applied (at higher loads/watts, the speed/pedaling frequency will be low and the exercise more static- than ‘explosive’-oriented). Additionally, an important consideration was ensuring that the players remained fresh and free from fatigue, as an extended warm-up could potentially diminish their energy levels and substantially change the baseline potential of the energy systems. Thus, we preferred a relatively short warm-up followed by lactate levels measurement to ensure that no athlete exceeded 3 mmol/L. Importantly, since all athletes underwent the same warm-up, we believe this allowed a valid and reliable comparison between groups regarding energy system contributions, which was the main goal of the study.
Comment 7:
When describing the results, it is not necessary to mention the statistical test. This is not a very common model! I recommend presenting the results and their respective significance values (p=)
Response 7:
Thank you for your suggestion. We understand the importance of clarity in presenting results and revised this section to focus on the results and their respective significance values, omitting the mention of specific statistical tests.
Comment 8:
Reference this part. But this issue needs more clarity in the writing. Because there are differences in the proportions of fiber types and this will be directly related to the individual's mechanical capacity, especially for short and supramaximal efforts. As well as their capacity for muscle recovery due to energy demand and metabolic efficiency in the administration of fatigue.
Response 8:
Thank you for your feedback. We have added the reference to this section and clarified the writing to better emphasize the differences in fiber type proportions and their direct impact on mechanical capacity.
“Sex differences in muscle fibers are evident. Men generally exhibit larger cross-sectional areas across all types of muscle fibers, with a greater percentage and area distribution for type II, IIA, and IIX fibers [41]. Conversely, women demonstrate greater distribution and area for type I fibers [41]. Additionally, while type II fibers contract at higher velocities and generate more force and power, type I fibers have a greater capacity for oxidative energy production and exhibit more endurance [42-44]”.
Comment 9:
Font color in red, modify.
Response 9:
Thank you, modified.
Comment 10:
The study is quite interesting. They discussed mechanical and physiological aspects very well. As well as biochemical issues. However, the discussions on the different characteristics of the athletes lacked a little more exploration. For example, endurance athletes tend to be better and more efficient in terms of the aerobic component, due to central (stroke volume, cardiac output and VO2max) and peripheral (mitochondrial density) issues. Speed athletes, on the other hand, tend to be more efficient at the intramuscular level (mainly because they have a greater demand for type IIa fibers, which are intermediate and more adaptive to fatigue). These points deserve attention in order to strengthen the discussion and the arguments between the findings.
Response 10:
Thank you for your valuable feedback and for highlighting the aspects of our study that you found interesting. We have briefly addressed these issues in our discussion to strengthen the arguments regarding our findings.
“Endurance athletes typically exhibit superior efficiency in aerobic components, attributed to central factors such as stroke volume, cardiac output (Jones et al. 2000), and VO2max (Bassett et al. 2000), as well as peripheral factors including mitochondrial density (Holoszy et al. 1984, Perry et al. 2010). In contrast, speed athletes generally demonstrate greater efficiency at the intramuscular level, primarily due to their higher reliance on type IIa fibers (Trappe et al. 2015), which are intermediate and exhibit enhanced adaptability to fatigue (Lievens et al. 2020)”.
Thank you for reviewing our manuscript. Detailed responses and the corresponding revisions, highlighted in track changes, can be found in the re-submitted files below.
Comment 1:
Please put references in the same bracket.
Response 1:
Thank you for your suggestion. All the multiple references have been put in the same bracket.
Comment 2:
In the introduction, it is recommended not to use references in this format. I suggest describing the subject and referencing it at the end.
Response 2:
Thank you for the feedback.
Done.
Comment 3:
There was a lack of description and justification of the reason for this study, demonstrating its importance. Emphasizing the gaps on the subject.
Response 3:
Thank you for your valuable feedback. We appreciate your observation and added a brief justification in the introduction.
“To our best knowledge, no one directly considered sex differences in energy system contribution during sprinting in the context of the specificity of the disciplines. Such data may help to determine whether the quality of training stimuli should be diversified by sex. In addition, coaches and sports scientists may be provided with insights into the specific athletic predispositions of male and female athletes and help them select appropriate sports discipline or specialty.”
Comment 4:
Please put this sample characteristic data into a table.
Response 4:
Thank you for the comment. The data have been moved to Table 1 in the Results section.
Comment 5:
I suggest putting the experimental design in illustrative format, in a figure.
Response 5:
Thank you for your suggestion. The experimental design has been presented in the new Figure 1.
Comment 6:
This is an erroneous pattern of tests applied in research. Because 5 minutes of warm-up (mainly at 25 to 50w) is not enough to physiologically activate the body to submit to stimuli, especially at high intensity.
Response 6:
Thank you for your feedback. We appreciate your valuable insights on the warm-up protocol.
Obviously, a warm-up of constant low-to-moderate intensity would be inappropriate. We based our protocol on the work of Beneke et al. (2002) and other authors who employed a similar approach before the Wingate test (Lovell et al. 2012, Doria et al. 2020). A key aspect of our warm-up was incorporating short sprints to replicate the specificity of the main task. Unlike Beneke, who recommended only two sprints of 3 seconds each, our athletes performed additional sprints up to 5 seconds, so the pre-adaptation was even more pronounced. When developing or warming up for maximal speed-power/velocity, a relatively low external load must be applied (at higher loads/watts, the speed/pedaling frequency will be low and the exercise more static- than ‘explosive’-oriented). Additionally, an important consideration was ensuring that the players remained fresh and free from fatigue, as an extended warm-up could potentially diminish their energy levels and substantially change the baseline potential of the energy systems. Thus, we preferred a relatively short warm-up followed by lactate levels measurement to ensure that no athlete exceeded 3 mmol/L. Importantly, since all athletes underwent the same warm-up, we believe this allowed a valid and reliable comparison between groups regarding energy system contributions, which was the main goal of the study.
Comment 7:
When describing the results, it is not necessary to mention the statistical test. This is not a very common model! I recommend presenting the results and their respective significance values (p=)
Response 7:
Thank you for your suggestion. We understand the importance of clarity in presenting results and revised this section to focus on the results and their respective significance values, omitting the mention of specific statistical tests.
Comment 8:
Reference this part. But this issue needs more clarity in the writing. Because there are differences in the proportions of fiber types and this will be directly related to the individual's mechanical capacity, especially for short and supramaximal efforts. As well as their capacity for muscle recovery due to energy demand and metabolic efficiency in the administration of fatigue.
Response 8:
Thank you for your feedback. We have added the reference to this section and clarified the writing to better emphasize the differences in fiber type proportions and their direct impact on mechanical capacity.
“Sex differences in muscle fibers are evident. Men generally exhibit larger cross-sectional areas across all types of muscle fibers, with a greater percentage and area distribution for type II, IIA, and IIX fibers [41]. Conversely, women demonstrate greater distribution and area for type I fibers [41]. Additionally, while type II fibers contract at higher velocities and generate more force and power, type I fibers have a greater capacity for oxidative energy production and exhibit more endurance [42-44]”.
Comment 9:
Font color in red, modify.
Response 9:
Thank you, modified.
Comment 10:
The study is quite interesting. They discussed mechanical and physiological aspects very well. As well as biochemical issues. However, the discussions on the different characteristics of the athletes lacked a little more exploration. For example, endurance athletes tend to be better and more efficient in terms of the aerobic component, due to central (stroke volume, cardiac output and VO2max) and peripheral (mitochondrial density) issues. Speed athletes, on the other hand, tend to be more efficient at the intramuscular level (mainly because they have a greater demand for type IIa fibers, which are intermediate and more adaptive to fatigue). These points deserve attention in order to strengthen the discussion and the arguments between the findings.
Response 10:
Thank you for your valuable feedback and for highlighting the aspects of our study that you found interesting. We have briefly addressed these issues in our discussion to strengthen the arguments regarding our findings.
“Endurance athletes typically exhibit superior efficiency in aerobic components, attributed to central factors such as stroke volume, cardiac output (Jones et al. 2000), and VO2max (Bassett et al. 2000), as well as peripheral factors including mitochondrial density (Holoszy et al. 1984, Perry et al. 2010). In contrast, speed athletes generally demonstrate greater efficiency at the intramuscular level, primarily due to their higher reliance on type IIa fibers (Trappe et al. 2015), which are intermediate and exhibit enhanced adaptability to fatigue (Lievens et al. 2020)”.
Thank you for reviewing our manuscript. Detailed responses and the corresponding revisions, highlighted in track changes, can be found in the re-submitted files below.
Comment 1:
Please put references in the same bracket.
Response 1:
Thank you for your suggestion. All the multiple references have been put in the same bracket.
Comment 2:
In the introduction, it is recommended not to use references in this format. I suggest describing the subject and referencing it at the end.
Response 2:
Thank you for the feedback.
Done.
Comment 3:
There was a lack of description and justification of the reason for this study, demonstrating its importance. Emphasizing the gaps on the subject.
Response 3:
Thank you for your valuable feedback. We appreciate your observation and added a brief justification in the introduction.
“To our best knowledge, no one directly considered sex differences in energy system contribution during sprinting in the context of the specificity of the disciplines. Such data may help to determine whether the quality of training stimuli should be diversified by sex. In addition, coaches and sports scientists may be provided with insights into the specific athletic predispositions of male and female athletes and help them select appropriate sports discipline or specialty.”
Comment 4:
Please put this sample characteristic data into a table.
Response 4:
Thank you for the comment. The data have been moved to Table 1 in the Results section.
Comment 5:
I suggest putting the experimental design in illustrative format, in a figure.
Response 5:
Thank you for your suggestion. The experimental design has been presented in the new Figure 1.
Comment 6:
This is an erroneous pattern of tests applied in research. Because 5 minutes of warm-up (mainly at 25 to 50w) is not enough to physiologically activate the body to submit to stimuli, especially at high intensity.
Response 6:
Thank you for your feedback. We appreciate your valuable insights on the warm-up protocol.
Obviously, a warm-up of constant low-to-moderate intensity would be inappropriate. We based our protocol on the work of Beneke et al. (2002) and other authors who employed a similar approach before the Wingate test (Lovell et al. 2012, Doria et al. 2020). A key aspect of our warm-up was incorporating short sprints to replicate the specificity of the main task. Unlike Beneke, who recommended only two sprints of 3 seconds each, our athletes performed additional sprints up to 5 seconds, so the pre-adaptation was even more pronounced. When developing or warming up for maximal speed-power/velocity, a relatively low external load must be applied (at higher loads/watts, the speed/pedaling frequency will be low and the exercise more static- than ‘explosive’-oriented). Additionally, an important consideration was ensuring that the players remained fresh and free from fatigue, as an extended warm-up could potentially diminish their energy levels and substantially change the baseline potential of the energy systems. Thus, we preferred a relatively short warm-up followed by lactate levels measurement to ensure that no athlete exceeded 3 mmol/L. Importantly, since all athletes underwent the same warm-up, we believe this allowed a valid and reliable comparison between groups regarding energy system contributions, which was the main goal of the study.
Comment 7:
When describing the results, it is not necessary to mention the statistical test. This is not a very common model! I recommend presenting the results and their respective significance values (p=)
Response 7:
Thank you for your suggestion. We understand the importance of clarity in presenting results and revised this section to focus on the results and their respective significance values, omitting the mention of specific statistical tests.
Comment 8:
Reference this part. But this issue needs more clarity in the writing. Because there are differences in the proportions of fiber types and this will be directly related to the individual's mechanical capacity, especially for short and supramaximal efforts. As well as their capacity for muscle recovery due to energy demand and metabolic efficiency in the administration of fatigue.
Response 8:
Thank you for your feedback. We have added the reference to this section and clarified the writing to better emphasize the differences in fiber type proportions and their direct impact on mechanical capacity.
“Sex differences in muscle fibers are evident. Men generally exhibit larger cross-sectional areas across all types of muscle fibers, with a greater percentage and area distribution for type II, IIA, and IIX fibers [41]. Conversely, women demonstrate greater distribution and area for type I fibers [41]. Additionally, while type II fibers contract at higher velocities and generate more force and power, type I fibers have a greater capacity for oxidative energy production and exhibit more endurance [42-44]”.
Comment 9:
Font color in red, modify.
Response 9:
Thank you, modified.
Comment 10:
The study is quite interesting. They discussed mechanical and physiological aspects very well. As well as biochemical issues. However, the discussions on the different characteristics of the athletes lacked a little more exploration. For example, endurance athletes tend to be better and more efficient in terms of the aerobic component, due to central (stroke volume, cardiac output and VO2max) and peripheral (mitochondrial density) issues. Speed athletes, on the other hand, tend to be more efficient at the intramuscular level (mainly because they have a greater demand for type IIa fibers, which are intermediate and more adaptive to fatigue). These points deserve attention in order to strengthen the discussion and the arguments between the findings.
Response 10:
Thank you for your valuable feedback and for highlighting the aspects of our study that you found interesting. We have briefly addressed these issues in our discussion to strengthen the arguments regarding our findings.
“Endurance athletes typically exhibit superior efficiency in aerobic components, attributed to central factors such as stroke volume, cardiac output (Jones et al. 2000), and VO2max (Bassett et al. 2000), as well as peripheral factors including mitochondrial density (Holoszy et al. 1984, Perry et al. 2010). In contrast, speed athletes generally demonstrate greater efficiency at the intramuscular level, primarily due to their higher reliance on type IIa fibers (Trappe et al. 2015), which are intermediate and exhibit enhanced adaptability to fatigue (Lievens et al. 2020)”.
Reviewer 2 Report (Previous Reviewer 3)
Comments and Suggestions for Authors
Introduction:
There are some relevant articles on this topic that were missed by authors... I strongly recommend to revise the literature, because its possible to find more on this topic... Thus, you could improve intro and discussion sections using references like:
https://link.springer.com/article/10.1007/s00421-014-3093-5
https://link.springer.com/article/10.1007/s00421-014-3086-4
https://link.springer.com/article/10.1007/s00421-019-04270-y
https://link.springer.com/article/10.1007/s004210050632
https://doi.org/10.1123/ijspp.2022-0411
- https://doi.org/10.1080/02640414.2016.1227079
Author Response
Comment 1
There are some relevant articles on this topic that were missed by authors... I strongly recommend to revise the literature, because its possible to find more on this topic... Thus, you could improve intro and discussion sections using references like:
https://link.springer.com/article/10.1007/s00421-014-3093-5
https://link.springer.com/article/10.1007/s00421-014-3086-4
https://link.springer.com/article/10.1007/s00421-019-04270-y
https://link.springer.com/article/10.1007/s004210050632
https://doi.org/10.1123/ijspp.2022-0411
https://doi.org/10.1080/02640414.2016.1227079
Response 1:
Thank you for your valuable feedback. We appreciate your suggestion to revisit the literature on this topic. We have thoroughly reviewed the additional references you mentioned and have incorporated several of them into the introduction and discussion sections.
Reviewer 3 Report (Previous Reviewer 2)
Comments and Suggestions for Authors
Thank you for your revision.
Comments on the Quality of English LanguageNone.
Author Response
Thank you.
This manuscript is a resubmission of an earlier submission. The following is a list of the peer review reports and author responses from that submission.
Round 1
Reviewer 1 Report
Comments and Suggestions for Authors
Introduction
Lines 31 to 44 discuss energy systems. However, from the way it has been described, it seems that there is a segmented order of activation. And this is not the case! Gastin (2001) describes this interaction (to a lesser extent in the ATP-CP system due to limited energy supply).
In addition, here is another reference that may help describe this concept.
Hargreaves, M., Spriet, L.L. Skeletal muscle energy metabolism during exercise. Nat Metab 2, 817–828 (2020). https://doi.org/10.1038/s42255-020-0251-4
Line 48
He started talking about sprint without a brief introduction to this type of stimulus. I suggest you follow this link before describing sprint.
Materials and Methods
Line 85
5 and 10 km are not considered long distances, but medium distances. I suggest modify.
Line 89
What sense do taekwondo athletes make of this kind of analysis? Even though they are athletes with a high-intensity (and dense) intermittent characteristic, there is no relation between their skills and cyclical stimuli, such as sprint.
Line 179
In statistics, with regard to the normality test, the Kolmorogov-Smirnov test would be correct, given that the sample n= is over 50.
Line 182
Also in statistics, the one way test was used inappropriately. In this case, it would have been a T-test, due to the fact that there were two groups and for the analysis of time. I suggest adjusting the statistic.
Results
Ok!
Discussion
In the discussion, they described the influence of gender and body composition well. However, they failed to discuss another important factor that is part of the energy mechanism, especially in high-intensity efforts. The type of fibers. There is a difference between sexes and also for sprinting performance, fiber typology is a factor that is directly linked to physiological and mechanical responses.
For example, type IIa and IIx fibers. Mainly type IIa, which have fast and intermediate characteristics (oxidative and glycolytic) and are important in controlling fatigue during supramaximal efforts. In addition, type IIx fibers will allow for greater muscle power responses and this will be decisive in the type of exercise applied in this study. We know that these intramuscular mechanisms have not been evaluated, but discussing them will make the argument for the findings even more robust.
Here are some references that may help you:
Haizlip KM, Harrison BC, Leinwand LA. Sex-based differences in skeletal muscle kinetics and fiber-type composition. Physiology (Bethesda). 2015 Jan;30(1):30-9. doi: 10.1152/physiol.00024.2014. PMID: 25559153; PMCID: PMC4285578.
Nuzzo J. L. (2024). Sex differences in skeletal muscle fiber types: A meta-analysis. Clinical anatomy (New York, N.Y.), 37(1), 81–91. https://doi.org/10.1002/ca.24091
O'Reilly J, Ono-Moore KD, Chintapalli SV, Rutkowsky JM, Tolentino T, Lloyd KCK, Olfert IM, Adams SH. Sex differences in skeletal muscle revealed through fiber type, capillarity, and transcriptomics profiling in mice. Physiol Rep. 2021 Sep;9(18):e15031. doi: 10.14814/phy2.15031. PMID: 34545692; PMCID: PMC8453262.
Finally, I suggest explaining the limitations of the study, future research and the importance of these findings in terms of practical applicability;
Author Response
Dear Reviewer,
Thank you for taking the time to review this manuscript. Below, you will find detailed responses and the corresponding revisions highlighted in track changes within the re-submitted files.
Comment 1:
Introduction
Lines 31 to 44 discuss energy systems. However, from the way it has been described, it seems that there is a segmented order of activation. And this is not the case! Gastin (2001) describes this interaction (to a lesser extent in the ATP-CP system due to limited energy supply).
In addition, here is another reference that may help describe this concept.
Hargreaves, M., Spriet, L.L. Skeletal muscle energy metabolism during exercise. Nat Metab 2, 817–828 (2020). https://doi.org/10.1038/s42255-020-0251-4
Response 1:
Thank you for your feedback and references regarding the definition of energy systems. We appreciate your interest in our paper and value your concerns. We agree with the reviewer and understand the reader may have an impression that the systems activate in a segmented manner. However, our goal was to present simultaneous activation, and we have revised the introduction accordingly.
Comment 2:
Line 48
He started talking about sprint without a brief introduction to this type of stimulus. I suggest you follow this link before describing sprint.
Response 2:
Thank you for this comment. We have added a brief introduction and believe this will enhance the readability of the manuscript.
Comment 3:
Materials and Methods
Line 85
5 and 10 km are not considered long distances, but medium distances. I suggest modify.
Response 3:
Thank you for your suggestion. However, according to World Athletics, distances are categorized as follows: sprints - up to 400m, middle/long distances - 800m to 1500m, including the 1-mile race, and distances of 3000 meters and longer. Specifically, 5km and 10km runs are considered classic long-distance track and field disciplines (see: https://worldathletics.org/disciplines/middlelong/5000-metres).
World Athletics groups middle/long distances performed on track (800m, 1500m, 3000m steeplechase, 5000m, and 10,000m) together, while separately listing road running (half marathon, marathon), cross country, mountain running, ultra running, and trail running (see: https://worldathletics.org/our-sport). Therefore, the categorization is more about race conditions than distance. It appears more crucial to classify distances based on physiological considerations and exercise responses, such as the predominant energy systems involved. A 'long' distance typically results in the overwhelming dominance of the aerobic system.
Duffield et al. (2003) showed the relationship between the percentage contribution of aerobic energy and distance duration (Figure 2), (see: https://centrostudilombardia.com/wp-content/uploads/IAAF-Corsa-Generale/2003-Energy-system-contribution-in-track-running.pdf).
After 300 to 350 seconds (6 to 7 minutes), the aerobic system contributes roughly 90%. Elite athletes typically finish a 5000m race in approximately 12 to 13 minutes, with women completing it in 15 to 16 minutes, resulting in notably higher aerobic system utilization, approximately 96-97%. This pattern is similarly observed in the 10,000m distance. These classifications are based on both practical usage and typical physiological responses.
Comment 4:
Line 89
What sense do taekwondo athletes make of this kind of analysis? Even though they are athletes with a high-intensity (and dense) intermittent characteristic, there is no relation between their skills and cyclical stimuli, such as sprint.
Response 4:
Thank you for your comments and observations.
The aim of the study was to observe the metabolic responses of athletes from various disciplines during a 15-second all-out test, considering their specific adaptations, rather than providing a comprehensive representation of the effort involved in each discipline.
Based on the characteristics of the discipline described by Craig et al. (2014) (DOI: 10.1007/s40279-014-0159-9), taekwondo athletes require muscular power, strength, and endurance to execute and maintain technical and tactical actions. Their training includes elements such as accelerationand bursts of effort during kicks, typically in the form of, cyclic movements.
Craig et al. (2014) indicated that the 30-second Wingate test is the most common method for assessing the peak anaerobic power and capacity of taekwondo competitors. Their review confirms the intense anaerobic nature of this combat sport and suggests that the ability of the lower limbs to generate high peak power may be crucial in competition.
Sadowski et al. (2012) (DOI: 10.12659/AOB.883279) showed that successful male taekwondo athletes (medalists in the Polish Senior Taekwondo Championships) achieved higher peak power on the Wingate test compared to their less successful counterparts (non-medalists in the Polish Senior Taekwondo Championships).The authors suggested that this difference could impact the effectiveness (speed and strength) of kicking techniques during bouts. Hence, generating high peak anaerobic power in the lower extremities may be crucial for competitive success.
Moreover, based on Janowski et al. (2019) (DOI: 10.1519/JSC.0000000000003110), following recent rule changes, taekwondo athletes are now required to engage in more rigorous anaerobic training. While our 15-second all-out test on the cycloergometer may not precisely mirror the specifics of the effort, the energy demands are expected to be comparable.
Additionally, our athletes participated in a familiarization session to acquaint themselves with the procedures and testing protocols to avoid mistakes due to a lack of cycling skills.
Comment 5:
Line 179
In statistics, with regard to the normality test, the Kolmorogov-Smirnov test would be correct, given that the sample n= is over 50.
Response 5:
Thank you for the suggestion. Nevertheless, we used the Shapiro-Wilk for assessing the normality of the distribution separately for separate subgroups (so, the numbers are much less than 50). Therefore, the Shapiro test was justified.
Comment 6:
Line 182
Also in statistics, the one way test was used inappropriately. In this case, it would have been a T-test, due to the fact that there were two groups and for the analysis of time. I suggest adjusting the statistic.
Response 6:
Thank you for your feedback. However, based on (Mishra et al. (2019) (doi: 10.4103/aca) "The one‑way ANOVA is an extension of independent samples t-test"
Thus, the p-values will be identical, and the F-test value (for ANOVA) will be equal to the t-test value squared (F=t²). Therefore, these two tests with two independent groups are interchangeable and yield identical results. Consequently, any correction in the manuscript would not alter the outcome or decision.
Comment 7:
Result
Ok!
Response 7:
Thank you for the positive feedback.
Comment 8:
Discussion
In the discussion, they described the influence of gender and body composition well. However, they failed to discuss another important factor that is part of the energy mechanism, especially in high-intensity efforts. The type of fibers. There is a difference between sexes and also for sprinting performance, fiber typology is a factor that is directly linked to physiological and mechanical responses.
For example, type IIa and IIx fibers. Mainly type IIa, which have fast and intermediate characteristics (oxidative and glycolytic) and are important in controlling fatigue during supramaximal efforts. In addition, type IIx fibers will allow for greater muscle power responses and this will be decisive in the type of exercise applied in this study. We know that these intramuscular mechanisms have not been evaluated, but discussing them will make the argument for the findings even more robust.
Here are some references that may help you:
Haizlip KM, Harrison BC, Leinwand LA. Sex-based differences in skeletal muscle kinetics and fiber-type composition. Physiology (Bethesda). 2015 Jan;30(1):30-9. doi: 10.1152/physiol.00024.2014. PMID: 25559153; PMCID: PMC4285578.
Nuzzo J. L. (2024). Sex differences in skeletal muscle fiber types: A meta-analysis. Clinical anatomy (New York, N.Y.), 37(1), 81–91. https://doi.org/10.1002/ca.24091
O'Reilly J, Ono-Moore KD, Chintapalli SV, Rutkowsky JM, Tolentino T, Lloyd KCK, Olfert IM, Adams SH. Sex differences in skeletal muscle revealed through fiber type, capillarity, and transcriptomics profiling in mice. Physiol Rep. 2021 Sep;9(18):e15031. doi: 10.14814/phy2.15031. PMID: 34545692; PMCID: PMC8453262.
Finally, I suggest explaining the limitations of the study, future research and the importance of these findings in terms of practical applicability;
Response 8:
Thank you for the feedback and the valuable references. While we agree that the issue of muscle fiber types is crucial in the context of energetics, we originally did not include it in our interpretation due to the non-invasive nature of our study. However, at the request of the reviewer, we have included an explanation in the discussion on the role of muscle fibers in the energetics of men and women. In addition, we have included the limitations of the study, future research directions, and the practical implications of our findings.
Reviewer 2 Report
Comments and Suggestions for Authors
Dear authors,
I am very glad to read this relevant work in the field of athletic training and energy source metabolism.
Considering the baseline difference of body mass between genders, the metabolism of fuel can be different intuitively. There are lots of variables can impact/alternate the energy expenditure, not limited to gender, age, body weight, body composition, sedentary behavior, athletic training modes, family history of certain metabolic syndromes, etc. However, it really surprised me that this study failed to observe significant difference between genders regarding PCr-Gycolytic-Aerobic system metabolism, but only found significant differences in body composition between genders.
The issue I would recommend to be noticed is that there shall be a section presenting the exiting limitations of this current study, including but not limited to the factors I mentioned above. Secondly, the difference between groups was performed using One-way ANOVA, while, I believe MANOVA, ANCOVA or step by step Factorial-ANOVA considering those above factors can be accessed if possible instead of using One-way ANOVA by itself.
Best regards,
Reviewer
Author Response
Dear Reviewer,
Thank you for dedicating your time to reviewing this manuscript. Enclosed, you will find detailed responses and corresponding revisions, clearly highlighted in track changes within the re-submitted files.
Comment 1:
I am very glad to read this relevant work in the field of athletic training and energy source metabolism.
Considering the baseline difference of body mass between genders, the metabolism of fuel can be different intuitively. There are lots of variables can impact/alternate the energy expenditure, not limited to gender, age, body weight, body composition, sedentary behavior, athletic training modes, family history of certain metabolic syndromes, etc. However, it really surprised me that this study failed to observe significant difference between genders regarding PCr-Gycolytic-Aerobic system metabolism, but only found significant differences in body composition between genders.
The issue I would recommend to be noticed is that there shall be a section presenting the exiting limitations of this current study, including but not limited to the factors I mentioned above. Secondly, the difference between groups was performed using One-way ANOVA, while, I believe MANOVA, ANCOVA or step by step Factorial-ANOVA considering those above factors can be accessed if possible instead of using One-way ANOVA by itself.
Best regards,
Response 1:
Thank you for your insightful review and valuable suggestions.
We recognize that numerous factors can influence energy metabolism in men and women. However, we have taken steps to control these variables. Our athletes were matched for age, weight, and body composition. Both groups were homogeneous in terms of training status and performance level (see Table 1), and pre-study questionnaires confirmed the absence of metabolic diseases among all subjects. We were also surprised to observe no significant sex differences in PCr-Glycolytic-Aerobic system metabolism in our study. This finding highlights the complexity of these interactions and is in line with other studies of Tortu et al. (2024) (doi: 10.16926/par.2024.12.02), Duffield et al. (2004) (doi:10.1016/s1440-2440(04)80025-2.), and Massini et al. (2021) (doi:10.3389/fphys.2021.796886.).
We acknowledge that our manuscript initially lacked a limitations section, and we have now included it. Please note that two-way analysis of variance (ANOVA) was conducted to examine the effects of athletic group, sex, and their interaction (Figures 1 and 2, Table 2) on main variables, according to our hypotheses. One-way ANOVA was only used to assess the magnitude of differences in accompanying characteristics between males and females within athletics groups (Figure 3).
Reviewer 3 Report
Comments and Suggestions for Authors
Although there are few studies exploring EPOC in single short supra-maximal exercises (and this was the reason I agreed to review this article), this study presents some significant methodological and statistical analysis constraints.
Specific comments:
#LINE 155: This equation is for on-kinetics, not for off-kinetics. Revise.
#LINE 160-174: The fit procedures should be defined a priori, not by convenience. What is the rationale for the Bertuzzi et al. argument to use only 6 minutes if you have 10? EPOC-related studies for short supra-maximal events suggest 10 minutes of recovery period.
Author Response
Dear Reviewer,
Thank you for taking the time to review this manuscript. Below, you will find detailed responses and the corresponding revisions, highlighted with track changes in the resubmitted files.
Comment 1:
Although there are few studies exploring EPOC in single short supra-maximal exercises (and this was the reason I agreed to review this article), this study presents some significant methodological and statistical analysis constraints.
#LINE 160-174: The fit procedures should be defined a priori, not by convenience. What is the rationale for the Bertuzzi et al. argument to use only 6 minutes if you have 10? EPOC-related studies for short supra-maximal events suggest 10 minutes of recovery period.
Response 1:
Thank you for your comments and feedback on the study. We appreciate your interest in our paper and value your concerns.
We recorded excess post-exercise oxygen consumption over 10 minutes, taking into account technical considerations to ensure the safety and accuracy of our subsequent analyses, particularly given the diverse breathing characteristics among our study groups (e.g., triathletes and sprinters). In our opinion, setting a predetermined time would not be appropriate and could make it difficult to properly fit the curve to the data (which is key in this method). In fact, there is no gold standard and EPOC times for analysis vary from study to study. The EPOC curve depends on exercise intensity, exercise time, and individual athlete’s response, the latter not fully predictable, hence we left some time reserve for analysis.
We focused extensively on analyzing the duration of EPOC to precisely model each kinetic curve. To achieve optimal fitting of each EPOC kinetic curve (r² > 0.80), we evaluated oxygen consumption calculations at different time intervals ranging from the 3rd to the 10th minute of recovery. Based on criteria of reliability and consistency, we centered our analysis on the 7-minute EPOC period. The chosen recovery time aligns with other studies, typically evaluated from the 6th to the 10th minute following high-intensity exercises such as the 100m sprint or 15- to 30-second all-out cycling efforts.
Here are some references:
Bertuzzi et al. 2016 (doi:10.1371/journal.pone.0145733)
To analyze the impact of recovery time on alactic metabolism calculations, authors estimated ALMET using EPOC data for 6, 8, and 10 minutes of recovery. The results showed no significant differences in ALMET calculations across these times. Therefore, GEADE-LaB users are advised to measure VO2 breath-by-breath at least during the first 6 minutes of recovery to determine ALMET.
Doria et al. 2020 (https://doi.org/10.1007/s00421-020-04392-8)
Pulmonary oxygen uptake (V̇ O2) was continuously measured during the Wingate test, at rest, during the warm-up, and 10 min of recovery.
Artioli et al. 2012 (DOI: 10.3791/3413)
“After collecting exercise oxygen consumption data, keep recording oxygen consumption for ten minutes before shutting the equipment down”.
Park et al. (2021) (doi:10.3390/biology10030198)
After 100-m sprint – 6 min recovery VO2
Comment 2:
Specific comments:
#LINE 155: This equation is for on-kinetics, not for off-kinetics. Revise.
Response 2:
Thank you for the comment. We based the procedures on Bertuzzi et al. (2016) (doi:10.1371/journal.pone.0145733) and Özyener et al. (2001) (doi:10.1111/j.1469-7793.2001.t01-1-00891.x.)
Additionally, according to Jones AM, Poole DC (Eds.), 2005 („Oxygen uptake kinetics in sport, exercise and medicine”. Routledge. Taylor and Francis Group, pages 24-26), the equation we implemented has been used correctly:
The equation for off-kinetics (bi-exponential (used in our study)
- V̇O2(t) = V̇O2baseline + Af[e-(t-td)/τf] + As[e-(t-td)/τs]
The equation for on-kinetics (bi-exponential) is as follows
- V̇O2(t) = V̇O2baseline + Af[1-e-(t-td)/τf] + As[1-e-(t-td)/τs]
Please note the key difference in the two equations. The equation 1) for off-kinetics contains 'e-', wheras the equation 2) for on-kinetics contains '1-e'.
Round 2
Reviewer 2 Report
Comments and Suggestions for Authors
Dear authors,
Thank you for your revision, according to my last letter, I suggested you make different quantitative synthesis regarding different variables and co-factors, e.g. ANCOVA, while, from the revised manuscript, there is no change in the statistical difference analysis.
Best regards,
Reviewer
Comments on the Quality of English LanguageNone